# Horse Transport to Three South American Horse Slaughterhouses: A Descriptive Study

**DOI:** 10.3390/ani10040602

**Published:** 2020-04-01

**Authors:** Béke Nivelle, Liesbeth Vermeulen, Sanne Van Beirendonck, Jos Van Thielen, Bert Driessen

**Affiliations:** 1Laboratory of Livestock Physiology, Department of Biosystems, KU Leuven, 3001 Heverlee, Belgium; bert.driessen@dierenwelzijn.eu; 2Dier&Welzijn vzw, 3583 Paal, Belgium; 3Westvlees NV, 8840 Westrozebeke, Belgium; liesbeth_vermeulen@westvlees.com; 4Bioengineering Technology TC, KU Leuven, 2440 Geel, Belgium; sanne.vanbeirendonck@kuleuven.be (S.V.B.); jos.vanthielen@kuleuven.be (J.V.T.); 5Thomas More, 2440 Geel, Belgium

**Keywords:** horses, transport, slaughter

## Abstract

**Simple Summary:**

In the western world, the number of slaughtered horses is decreasing, but still about 5 million horses are slaughtered worldwide each year. The conditions in which horses are transported to the slaughterhouses are a topic of discussion. This study intended to investigate the circumstances of commercial slaughter horse transport and to detect possible risk factors for horse welfare. Therefore, 23 commercial horse transports to three South American slaughterhouses were monitored. During transport, a camera was mounted in each loading space so that horse behaviour could be analysed after transport. Fighting behaviour could not be explained by stocking density, environmental parameters, trailer characteristics, duration and distance of the journey. The temperature and relative humidity were recorded every five minutes in all loading spaces. Average temperatures exceeded the thermoneutral zone in six transports, but it is not clear if and to what extent horse welfare was impaired. Overall, loading and transporting of the horses went well, but the infrastructure of the loading area did not always promote smooth loading and can therefore be improved. At later visits, we noted that this issue was addressed.

**Abstract:**

Between November 2016 and October 2017, 23 horse transports from 18 collection points to two slaughterhouses in Argentina and one in Uruguay were monitored. The goal of this study was to characterize the current practices in commercial horse transports and to detect potential threats to horse welfare. A total of 596 horses were transported over an average distance of 295 ± 250 km. Average transport duration was 294 ± 153 min. The infrastructure did not always promote smooth loading, but the amount of horses that refused to enter the trailers was limited. In each loading space, a camera was mounted to observe horse behaviour during the journey. Ambient temperature and relative humidity (RH) were recorded every five minutes in each loading space. In 14 of the 23 transports, the maximum temperature rose above 25 °C and the average temperature was over 25 °C during six transports. The average temperature humidity index (THI) exceeded 72 during six transports. The average stocking density was 1.40 ± 0.33 m^2^ per horse, or 308 ± 53 kg/m^2^. The degree of aggression differed between the front and rear loading space. Stocking density, environmental parameters, trailer characteristics, and transport duration and distance did not influence aggressiveness.

## 1. Introduction

In the USA, Australia and Europe, the number of slaughtered horses is decreasing. In Belgium for example, the number of slaughtered horses decreased from 21,390 slaughtered horses in 2001 to only 5895 horses in 2018 [1]. Annually, about 5 million horses are slaughtered worldwide [2]. In 2018, China was the country that slaughtered the most horses worldwide—more specifically, about 1.59 million [2]. In the same year, 389,153 horses were slaughtered in South America [2]. However, to this day, the circumstances in which horses are transported give rise to discussion [3].

Animals should be handled as carefully as possible at all times, including during the transportation process to the slaughterhouse. The animals should undergo as little stress as possible, on the one hand for welfare reasons, but also to prevent any deterioration in the quality of the horse meat. Quality loss can occur as a result of excessive stress, bruises or injuries [4,5,6,7]. Suboptimal ambient parameters such as temperature, relative humidity, ammonia and carbon dioxide concentration in the air can cause stress, but also inappropriate infrastructure and psychological stressors. Examples of psychological stressors are the determination of dominance rank, and transport activities such as loading, unloading and the transport itself. In addition, activities that are part of the management of the animals, such as the weaning of young animals, weighing of animals or changing housing, can also cause stress [8].

The conditions of (non-commercial) transports of sport and company horses and the response of these horses to transport are well studied [9,10]. On the other hand, commercial transport of horses is studied to a lesser extent. Furthermore, most studies in which horses are transported untethered in groups involve healthy horses that are used to being transported [9,11,12,13,14]. Studies investigating slaughter horse transports or horses that were sold as slaughter horses are less numerous [13,15,16,17,18,19]. However, from those studies it is clear that a number of transport-related factors influence horse welfare. Journey distance and time [20,21], loading density [7,15,22,23], handling [7,24,25], new environments [24], potentially re-grouping or mixing with unfamiliar animals [24], fasting and deprivation of water [24], the myriad of trailer designs [7,23], driving behaviour [7], road type and quality [7,22,24], traffic conditions [24], suspension systems and building materials of the trailers [15,23], environmental conditions in the trailer [7,22], and weather conditions [15,23,26] all affect horse welfare.

According to Morgan [27], the thermoneutral zone of a horse is on average between 5 °C and 25 °C. Another study defines the thermoneutral zone between −1 °C and 24 °C [27]. The differences in estimation of the thermoneutral zone are, among other things, probably due to acclimatization, body condition and climate [28]. For example, the upper limit of the thermoneutral zone lowers with increasing humidity. At a relative humidity (RH) of more than 50%, it gets harder for the animals to dissipate heat to the environment [29]. The temperature humidity index (THI) is a useful parameter to estimate the thermal comfort of organisms [30].

Legislation involving animal welfare differs between countries. At the same time, meat is traded from countries with less stringent welfare requirements, like Argentina [31,32,33] and Uruguay [34], to countries with higher welfare requirements, like members of the European Union. The European Union (EU) sets welfare requirements at the time of killing for companies willing to export animal products to the EU (Council Regulation (EC) No 1099/2009) [35], but does not impose direct requirements on the transport of those animals to the slaughterhouses in third countries. The international Horse Meat Federation [36], on the other hand, expects its members to meet the requirements set in their “Manual for the Animal Welfare of horses during transport and slaughtering”, which is based on existing legislation and international guidelines [36].

The first aim of this study was to characterize the current practices of the commercial horse transport from collection points to slaughterhouses in Argentina and Uruguay. Secondly, potential risk factors for horse welfare were detected.

## 2. Materials and Methods

Between November 2016 and October 2017, a total of 23 horse transports in Argentina and Uruguay were monitored from loading at a collection point to one of the three selected horse slaughterhouses. A total of 596 half-bred horses with an average weight of 415 (±38) kg were picked up at 18 collection points and were transported to Lamar (Argentina), Frigorífico General Pico (Argentina) and Sarel (Uruguay). The transports were spread throughout a year, so that a number of transports were monitored in each season. A total of six transports took place in the spring (November 2016), six in the summer (March 2017), six in the fall (June 2017) and five in the winter (October 2017). Each season two transports per slaughterhouse were monitored, except for October 2017. At that time, only one transport to Frigorífico General Pico was followed up. The transports were carried out with different types of transport vehicles. In this study, three types of vehicles are distinguished, namely trucks, tractor–trailers and truck–trailers (Figure 1). A truck is a pulling vehicle with one inseparable loading space and is therefore a single unit. In the case of a tractor–trailer, the towing vehicle can be (dis)connected from the trailer via a fifth-wheel coupling. Finally, a truck–trailer is a truck, as defined above, with a trailer connected to it through a drawbar. This transport combination therefore exists of two separated loading spaces.

The transports were monitored and supervised by the same researcher. The researcher also mounted the sensors and cameras in the trailers and recorded specific transport data. In each trailer the horses were filmed. The camera (Trophy Cam model 119437, Bushnell, China) filmed fragments of one minute at intervals of about 100 seconds. On average, 38.9% (±3.8)% of each transport was filmed. After the transports, the footage was viewed, and the behaviour and interactions of the horses were analysed. During the analysis of the videos, it was noted per trailer how many horses fell and whether the animals fought ‘hardly or not’, ‘averagely’ or ‘a lot’. A fall was considered to be a loss of balance in which parts of the body other than the hooves touched the ground. If the horse could restore equilibrium without other body parts touching the ground, this was called stumbling. Furthermore, the temperature, relative humidity (RH) and dew point were automatically recorded (EL-USB-2, Lascar Electronics, Wiltshire, UK) every five minutes in each loading space. These data were automatically written to an excel file. For the analysis of the environmental parameters, the thermoneutral zone used by Morgan [27], namely between 5 °C and 25 °C, is taken as the starting point. In addition, the temperature humidity index (THI) is calculated using the following formula [37], with T, temperature in °C, RH expressed as a number between 0 and 1:(1)$$\mathrm{THI}=0.8T+\mathrm{RH}\times \left(T-14.4\right)+46.4$$

The timing of various operations such as loading and unloading, the duration and distance of the transport, the number of intermediate stops for (police) checks, as well as the stocking density, dimensions and characteristics of the loading spaces were recorded. The observer noted what tools the drivers, which are the persons that handle the horses, used. Furthermore, the characteristics and dimensions of loading docks were registered. The openness of the side walls was categorised as ‘open’ when the surroundings could be seen easily through the wall, ‘half open’ when the view through the side walls was limited, and ‘solid’ when the horses could not see anything through the side walls of the loading dock. All the parameters that were considered during this investigation are listed in the Appendix A in Table A1 and Table A2.

The data were processed using SAS Enterprise Guide and SAS 9.4. Averages and standard deviations were calculated using the PROC UNIVARIATE procedure. Correlations between the environmental parameters in the front and rear trailer were calculated using the regression (PROC REG) procedure. The frequency procedure (PROC FREQ) was used for all frequency calculations and generalized linear mixed models (PROC GLIMMIX) were used to identify which parameters influenced the degree of aggression and falling of horses during transport.

## 3. Results

### 3.1. Loading Dock

In 17 of the 18 collection points, a loading dock was present to facilitate the loading of the horses. From two collection points, horses were picked up more than once. However, only one loading dock was used for two transports. In one collection point, the loading dock did not have a slope, since the transport vehicle could be parked so that surface of the loading area was at the same level as the floor of the trailer. Therefore, this loading deck was excluded from the averages (Table 1). The slope of the loading docks was on average 17.4° (±3.6)°, which equals 31.5% (±7.0)% (Table 1). Seven out of 21 loading docks had a slope steeper than 20.0° (36.4%) and the slope of all but one loading dock was steeper than 10.0° (17.6%). The length of the loading dock (measured on the surface of the loading dock) was 4.01 (±0.90) m and the height was 1.18 (±0.17) m (Table 1).

The surface of the loading docks consisted of only soil (26.1%), a combination of soil and wood (26.1%), only wood (13.0%), concrete (partly) covered with soil (8.7%), a combination of soil and grit (8.7%) or straw (4.4%). The side walls of the loading docks were, on average, 1.59 (±0.22)-m-high and constructed from wood—mostly planks (82.6%), but round wooden beams (4.3%) in one instance. The side walls of three (13.0%) loading docks were categorised as ‘open’, seven (30.4%) as ‘half open’ and 10 (43.5%) as ‘solid’. Of three (13.0%) loading docks, the kind of side wall construction was not registered.

### 3.2. Loading

Loading the horses took an average of 12.2 (±7.1) minutes per transport and 0.49 (±0.27) minutes per horse (Table 2). Spread over three transports, five horses (0.84%) had to be led into the trailer with a halter: three horses in one loading did not want to enter the trailer and twice one horse refused to enter the trailer. In the end, two of these horses could not be loaded at all. The trucks left the collection points 8.3 (±4.4) minutes after the loading process was completed. In total, 73.9% of the transports departed in the morning, on average at 11:15 a.m. (±1:31; between 9:25 a.m. and 3:20 p.m.). Figure 2 shows the arrival and departure times of the transports.

Tools such as a flag, whip or stick were used to drive the horses on the trailers. We define a stick as a narrow, long and little or not flexible object to drive the horses with. A whip is a thin, not very flexible stick, with or without a handle on it. In this context, a flag is a stick with a piece or ribbons of textile or plastic, so that the movements of the object are more visible to the horses. Flags were used in 22 (95.7%) of the 23 transports. Four times (17.4%) a whip or a stick was used and three times (13.0%) a rider on horseback drove the slaughter horses onto the trailer. The tools were not used to hit or poke animals on sensitive body parts, but to give visual signals. Occasionally, the horses were gently touched with the stick, whip or flag, but not to the extent that the touch could cause pain or discomfort.

Divided over 13 loads, a total of 23 (3.86%) horses stumbled during loading, with a maximum of five horses during one loading. Five falling horses were noted, spread over three loadings. During a loading in which three horses stumbled, also three horses fell and during two other loads, in which respectively three and five horses stumbled, one horse fell each time. No falls were noted in other loadings. The falls were caused by pushing and/or fights among horses. One of the fallen horses was ran over by the other horses when the group abruptly turned around in the narrowing space towards the loading ramp. The abrupt turn of the group was caused by the directions of the drivers.

Horses enter the transport vehicles through trapdoors, which are guillotine-type doors. Internal doors were also trapdoors. During 13 of the 23 transports, at least one horse bumped its head against the trapdoor. In five transports, three or more horses (maximum eight) bumped their heads against the trapdoor. The height of the trapdoors where horses hit their heads varied between 1.51 and 1.81 m.

### 3.3. Environmental Parameters

The lowest trailer temperature observed during transport was 6.0 °C, while the maximum temperature was 35.5 °C (Table 3). In the trailers, the temperature never dropped below the lower limit of the thermoneutral zone, being 5.0 °C. The upper limit of the thermoneutral zone, 25.0 °C, was exceeded during 14 of the 23 (60.9%) transports. During six (26.1%) transports, the average temperature in the trailer was above 25.0 °C and during five (21.7%) transports, the minimum temperature in the trailer exceeded 25.0 °C. For seven (30.4%) transports, the maximum temperature was above or equal to 30.0 °C. There was a strong correlation between the front and rear load for both average, minimum and maximum temperatures (Table 4).

The RH varied between 28.0% and 99.0% (Table 3). There was a strong correlation between the front and rear loading space for both average, minimum and maximum humidity (Table 4). There was no correlation between temperature and RH within the same loading space.

Due to the lack of a reference framework with limit values for heat stress in horses [38,39], the THI is tested against the values used for dairy cattle. A THI of 72–78 is labelled as mild heat stress, while a THI between 79 and 89 stands for severe heat stress in dairy cattle [40,41,42] (Appendix A, Figure A1). The THI ranged from 45.5–83.0 during transports (Table 3). In 13 (57%) transports, the maximum THI exceeded 72, which is the lower limit for mild heat stress in cattle. For six (26%) transports, the average THI was above or equal to 72 and for four (17.4%) transports, the minimum THI was at least equal to 72. During six (26.1%) transports, the maximum THI value was between 78 and 89, the standard for severe heat stress in cattle. However, the average THI value always remained below 78. There was a strong correlation between the front and rear loading space for both average, minimum and maximum humidity (Table 4). The average temperature, RH, dew point temperature and THI are shown per transport in Appendix A
Table A4.

### 3.4. Trucks

Different types of transport vehicles were used to carry out the 23 transports (see Materials and Methods). Some vehicles were used for multiple transports. Table A3 in the Appendix A shows the type of vehicle used for each transport and the frequency of use of the vehicle during the monitoring. Only one of the trucks had a roof consisting of a black sail. The average dimensions of the loading spaces are shown in Table A3 in the Appendix A. In 10 transports (43.5%) the front loading space was divided into several compartments: eight times (34.8%) into two compartments and twice (39.1%) into three. The rear loading space was divided into two (six times; 40.0%) or three (two times; 13.3%) compartments in eight (53.3%) of the 15 transports with two loading spaces.

The floor in all loading spaces was provided with wire mesh to prevent slipping of the horses. Different types of wire mesh could be distinguished. The most common were the standard wire mesh (Figure 3a), where the rods are on top of each other and the rods do not bend between the crossings. This type of wire mesh was used in 20 (87.0%) transports. The curved wire mesh (Figure 3b), which is bent between the crossings, was used in two transports, just like the diamond-shaped wire mesh with connections in one plane (Figure 3c). In the Appendix A
Table A3 shows which type of wire mesh was found per transport and per trailer. The average mesh size was 26.0 ± 4.5 cm by 23.7 ± 4.8 cm. On average, the wire mesh was 1.29 ± 0.38 cm thick and mounted at a height of 2.60 ± 0.78 cm. In our observations, both on the spot and afterwards, no shoed horses were detected in any of the transports.

### 3.5. Trailer Density

The average density of the trailers was 1.40 (±0.33) m^2^/horse or 308 (±53) kg/m^2^. Table 5 shows the average density in m^2^/horse and kg/m^2^ per trailer and per compartment. Average density varied between 0.94 m^2^/horse and 2.45 m^2^/horse. Stallions were not always separated from mares and geldings during transport. During at least five transports, one or more stallions were loaded. In at least two of these transports, the stallions were not separated from the other horses. Once, a stallion standing between mares was moved to another compartment before departure, because of his aggressive behaviour. In the other compartment however, the stallion was not separated from the other horses behind a fence or ropes either. To prevent further biting, a rope was tied tightly in the mouth. The stallion then stopped his aggressive behaviour. Several times a pony or a young horse was transported in the same compartment with significantly larger horses.

### 3.6. Aggression and Falling during Transport

The degree of aggression was assessed per loading space during 22 transports, of which 15 transports with a truck–trailer, together accounting for 38 loading spaces. A loading space refers to the space in one transport component. A truck therefore has one loading space just like a tractor–trailer, while a truck–trailer has two loading spaces. In 13 (34.2%) loading spaces, hardly any or no fights were registered. In 15 (39.5%) loading spaces, an average amount of fights was recorded, and a lot of fights were recorded in 11 (28.9%) loading spaces. No relation was found between the degree of aggression and the density (front loading space: *p* = 0.78; rear loading space: *p* = 0.25), the thickness of the wire mesh on the floor (front loading space: *p* = 0.23; rear loading space: *p* = 0.20), the transport duration (front loading space: *p* = 0.90; rear loading space: *p* = 0.98) or distance (front loading space: *p* = 0.93; rear loading space: *p* = 0.78), the average temperature (front loading space: *p* = 0.33; rear loading space: *p* = 0.79), the average dew point temperature (front loading space: *p* = 0.18; rear loading space: *p* = 0.99) and the average THI (front loading space: *p* = 0.28; rear loading space: *p* = 0.80). However, the degree of aggression differed between the front and rear loading spaces (*p* = 0.05). Splitting the rear loading space into two or more compartments resulted in less fighting than when the rear loadings spaces consisted of one compartment (*p* = 0.05). In the case of the front loading space (22 transports), no relation was found between compartmentalisation and the degree of fighting (*p* = 0.73). It should be noted, however, that the fighting behaviour in the rear loading space could only be monitored for 14 transports, since only 14 out of 15 truck–trailers had the rear loading spaces successfully filmed.

In only one journey did a horse fall twice. In the 22 other journeys, no horses fell, except once before departure. That horse was then removed from the truck and not taken to the slaughterhouse. No connection could be found between the degree of fighting and the falls of the horses, but this is likely due to the limited number of horses that fell during the journey.

### 3.7. Transport Distance and Duration

The transport distances from collection point to slaughterhouse ranged from 37–700 km (Table 6). The transport time varied from 73–632 min (Table 6). One transport lasted 480 min or eight hours, while two (8.7%) other transports lasted longer, more specifically 500 and 632 min. The horses were not unloaded during the journey and had no ability to eat or drink. The transport duration and distance were strongly correlated (r^2^ = 0.94; *p* < 0.0001) for the 23 observed transports: with increasing distance, the transport duration increased according to:(2)Y=49.7+0.8X
(with Y = the transport duration in minutes and X = the distance in km)

An average of 2.74 (±1.33) stops were inserted per transport, of which an average of 0.87 (±1.00) were inserted for police checks (Table 6).

### 3.8. Unloading

After arrival, transporters had to wait for an average of 14.8 (±12.6) minutes before unloading (Table 6). The unloading of the horses took on average 12.5 (±9.5) min per transport and 0.473 (±0.285) per horse (Table 6). There was no correlation between the duration of loading and unloading (r^2^ = 0.11; *p =* 0.08).

## 4. Discussion

### 4.1. Loading Docks

Besides good handling, the professional federation of the international Horse Meat Sector (HoMeFe) [36] as well as the European Consortium of the Animal Transport Guides Project (CATGP) [7] stresses the importance of good loading dock design, construction and maintenance to minimize the risk of slipping, falling, injuries and stress to animals while (un)loading. Therefore, CATGP [7] sums up a number of ‘good’ and ‘better’ practices in its guidelines for transport of slaughter horses. The good practices are derived from the Council Regulation 1/2005 of 22 December 2004 on the protection of animals during transport and related operations [7,43].

The loading area should be constructed in a way that prevents distress, excitement and injury as much as possible [7,36,43]. Good practices include that the slope of the loading dock does not exceed 36.4% (or 20.0°) [7,36,43,44], better practices demand a maximum slope of no more than 10.0° (17.3%) [7]. When the slope is steeper than 10.0° or 17.6%, the loading dock must be equipped with some sort of system that improves passage of animals without the risk of slipping [36,43], such as stair steps or foot battens [7,36,45]. Argentinian legislation states that the slope of the loading dock should not exceed 30.0° (57.7%) and should be equipped with foot battens [33]. Good loading practices also include using a slip-resistant and anti-sliding surface on the loading ramp [7,36,43]. To prevent animals from falling off or escaping from the (un)loading dock, side walls should be provided [7,36,43]. Side walls which limit the view on the environment prevent animals from being distracted by what is happening around the loading dock and might thereby simplify (un)loading [7,36,44]. Grandin [46] also mentions that solid side walls are more efficient in preventing escape attempts due to the blocked vision of the animals.

### 4.2. Loading and Unloading

Loading duration is considered to be an indicator of the ease of loading. Since no loading times of slaughter horses are available for comparison, the observed loading durations can only be compared to these of beef cattle. María et al. [47] noted an average loading time of 1.20 (±0.86) minutes per beef bull. However, horses and cattle cannot be compared in terms of the leniency of their movements; these data support the observation that the loading of the horses in general went quite smoothly.

However, infrastructure did not always promote smooth loading. Horses regularly bumped their heads against trapdoors, clearly indicating that the trapdoors were too low. Furthermore, when stressed, for example due to rushed driving, horses carry their head higher [48] and thereby the chance of head-bumping is increased, especially when trapdoor height is rather low. Argentinian legislations states that the trapdoors should be at least 1.60 m high, which was not always the case, to prevent the horses from hurting their heads and backs [33].

In a total of seven loadings, the observing researcher noted that horses hesitated or reacted anxiously to specific elements of the infrastructure; for example, low or not fully opened trapdoors, a bar hanging too low over the passageway, a steep slope of a loading dock, or an uneven or muddy ground on or before the loading dock. Falling horses were only observed in loadings in which multiple horses stumbled. The falls were caused by pushing or fights between horses, which was clearly caused by the directions of the drivers in one instance. The directions of the drivers were not necessarily wrong in this case, but with a calmer approach the fall might not have happened. Furthermore, drivers must ensure that they do not give conflicting signals to the horses, for example by standing too close to the passageway of the horses when another driver is directing the horses to go there. These conflicting signals create confusion and thereby chaos. However, the presence of the investigators was probably—to a certain extent—an additional stress factor for the horses and loading crew.

For all transports, unloading went quite well. However, better communication between the slaughterhouse and the transporter may reduce the standstill before unloading the horses on arrival.

According to Friend [23], all horses experience stress during transport. Loading and unloading might even be more stressful to animals than the transport itself [47], but horses that were loaded before and did not have any negative experiences with loading experienced less stress than animals that were loaded for the first time [49]. It is not known whether the horses observed in this study had been loaded and transported before, and how any previous transports were perceived. It is possible that the horses that did not want to enter the trailer had had previously negative experiences during a transport, or had other bad experiences [13]. If the person who loads the horses has a good relationship with the horses, the stress level during loading decreases [49]. Since the horses do not have much contact with people during rearing, there is little evidence that there is a relationship of trust between the caretakers and the horses. Due to the lack of (positive) transport experiences of the horses and the absence of a trust relationship with the drivers, the importance of efficient and knowledgeable driving increases, in order to make loading as smooth as possible [49]. The authors of [47] state that education and training of the personnel is likely to be one of the most effective measures to improve horse welfare during loading and transport. HoMeFe demands that everyone involved in the transport of slaughter horses is trained regularly [36]. However, it is not clear whether or not the drivers and horse transporters of the observed transports had training in the handling and transporting of horses.

HoMeFe prohibits the use of electric driving aids, sticks and dogs [36]. As well as the World Organisation for Animal Health (OIE) [50], Grandin [51] mentions that the use of an electric prod should be avoided if possible, since the electric prod might cause the animals to become agitated and therefore sometimes dangerous. Furthermore, drivers should not scream, flap their arms or make sudden movements to keep the animals as calm as possible [36,50,51]. After all, besides their adverse effects on animal welfare, multiple studies demonstrate the adverse effects of the use of electric prods and incompetent handling on meat quality in pork and beef [52,53,54].

No electric prods were used and handlers stayed calm. Flags were used in all but one loading. Against the requirements of HoMeFe [36], a whip or stick was used in four loadings, but never to hit horses.

### 4.3. Environmental Parameters

About three-quarters of the transports left before noon, implying that the transports were often being carried out during the hottest moments of the day.

When the environmental temperature rises above the upper limit of the thermoneutral zone, the animal has to invest energy to keep its body temperature constant [7,26]; for example, by sweating, peripheral vasodilation, and by increasing the respiratory rate [7,26]. However, it is not clear from what point welfare is compromised. The temperature regularly rose above 25 °C, which is the upper limit of the thermoneutral zone of horses [27], and RH regularly increased above 50%. Above 50%, heat is dissipated less efficiently [29].

Based on the THI framework for producing dairy cows (Appendix A
Figure A1) [42], mild and sometimes severe heat stress occurred during transports. However, THI must be interpreted carefully, since this parameter does not take solar load and wind speed into account [55]. Especially during transport in an open trailer, air displacement might enhance heat dissipation. On the other hand, the horses are standing close to each other during transport, which might limit heat dissipation. Moreover, the lack of a reference framework with limit values for heat stress in horses complicates interpretation.

Older horses are known to be more prone to heat stress than younger horses. When exposed to the same level of exercise, older horses overheat in a much shorter time than younger horses, indicating that their ability to dissipate excess heat is less compared to that of younger horses [56]. However, the age of the slaughter horses in this study was not known.

### 4.4. Trucks and Trailer Density

The floor of all transport vehicles was provided with wire mesh, as demanded by Argentinian legislation [33]. Different authorities and guidelines have determined the minimum and maximum density for horse transports, which explains the differences in thresholds. According to the CATGP [7], the density may vary between 1.00 m^2^ per horse and 1.75 m^2^ per horse, depending on the size and age of the horse (Appendix A
Table A5). The available space per horse may deviate by a maximum of 10% from the directive, depending on the physical condition, the weather conditions and the probable transport time [7]. HoMeFe [36] has drafted guidelines in its specifications for the density during transport. The available space must be between 1.1 m^2^ per horse and 1.4 m^2^ per horse. Depending on the physical condition of the horses, the weather conditions, the travel time, the weight and the height of the horses, the actual density may deviate a maximum of 20% from the guidelines. This means that the available surface area per horse may vary between 0.88 m^2^ and 1.68 m^2^ [36]. In one or more compartments of six transports, the average surface area per animal deviated more than 20% from the HoMeFe guideline [36]: the surface area per animal was too large in five instances, and it was too small in one instance (Appendix A
Table A6). It should be noted that the allowed stocking densities differ substantially between the CATGP [7] and HoMeFe [36] guidelines. However, comparing stocking densities expressed as surface area per horse is not evident, since adult horses can differ substantially in size. Two trailers with the same loading density expressed in m^2^/horse can be, in reality, quite a different stocking density in kg/m^2^ for these two loadings. On the other hand, determining the number of horses that can be loaded based on the estimated average weight of a group of horses might be prone to estimation errors.

Stull [57] compared some physiological parameters and the increase in injuries between horses transported at low (1.40–1.54 m^2^/horse) and high stocking densities (1.14–1.31 m^2^/horse). She concluded that it is better to provide at least 1.40 m^2^/horse during transport, depending on the weight, conformation and size of the horses. Extra attention must be paid to the design of the trailer in order to prevent injuries to the horses [57].

A few times a pony or a young horse was transported in the same compartment with other, significantly larger horses. This is contrary to European Regulation EC 1/2005 [43] and the Argentinian Resolución 581/2014 [33], which states that animals of significantly different sizes or ages and sexually mature mares and stallions, must be handled and transported separately, unless the animals have been reared together, are accustomed to each other, or when the separation would cause distress. Similarly, mares accompanied by their dependent foals are not subject the above provisions. Finally, animals ’hostile to each other’ and tied and untied animals should not be transported in the same compartment [43]. In at least five transports, one or more stallions were loaded. The stallions were not separated from the other horses in at least two transports, which is contrary to the abovementioned European Regulation [43], Argentinian legislation [33], and guidelines from HoMeFe [36]. One stallion that was not separated from the other horses was aggressive and was therefore placed in another compartment and had a rope tied tightly in the mouth. After putting on the rope, the stallion stopped behaving aggressively, but this was probably due to the inconvenience caused by the rope.

### 4.5. Aggression and Falling during Transport

In this study, no relationships could be demonstrated between environmental parameters and aggression. This suggests that the environmental parameters are not the most important factors that may or may not provoke aggression. As well as the current study (Appendix A
Table A6), Iacono and colleagues [18] could not demonstrate a relationship between the degree of aggression and the density or fatigue of the horses during the transport of untied horses. According to the authors of [18], aggression is more likely to be a consequence of individual horses than stocking density. The current observations confirm this assumption (Appendix A
Table A6).

### 4.6. Transport Distance and Duration

Argentinian legislation allows for a transport duration up to 36 hours without feeding, watering or rest [31]. In Uruguayan law, no maximum transport duration is mentioned [34]. According to the specifications of HoMeFe [36], a transport may take up to 12 h. In exceptional cases, the transport may last up to 14 h, for example if the destination can be reached by continuing the ride for a maximum of two hours [36]. This requirement differs substantially from the specifications of the European Regulation EC 1/2005 [43], that states that a transport may only take up to eight hours. In exceptional cases, the transport of trained horses may last up to 24 h if they are watered and fed every eight hours and if the transport vehicle meets some extra requirements for roof construction, presence and quality of litter, feeding and watering regime, partitions, ventilation, climate control and navigation system. However, for unbroken horses, the European Regulation does not allow any extension of the transport duration of eight hours, regardless of the transport vehicle in which the unbroken horses are transported [43].

Friend [14] claims that it is advisable to regularly provide the horses with water on the truck during long-distance transport in warm conditions, in order to reduce dehydration, stress and fatigue. CATGP [7] recommends to water horses every 4.5 h, while the European Council Regulation EC 1/2005 [43] states that Domestic Equidae have to be watered every eight hours. Notwithstanding, Friend [14] also mentions that water consumption can be highly variable among different horses in the same situation [11]. Likewise, other studies question the watering of horses during transport [17,58]. After all, it often takes some time before the horses start drinking, about 20 min to an hour after the water is offered. Some horses did not drink, possibly for fear of the new water source or because of stress associated with the transport. Furthermore, the difference in weight loss between horses that did and did not drink suggested that the horses probably did not drink a large amount of water [17,59]. Therefore, it seems especially important that the horses are sufficiently hydrated before departure and have access to sufficient fresh water immediately after arrival [14,36,60].

## 5. Conclusions

Our study identified the current practices of the commercial horse transport from collection points to slaughterhouses in Argentina and Uruguay. Some risk factors have been detected and could be improved. The loading and unloading of the horses generally went quite smoothly. Better training of drivers and optimized infrastructure (a level ground surface before and on the loading dock, sufficiently high trapdoors, steepness of the loading docks, provision of steps or foot battens on the loading dock, etc.) can prevent a lot of confusion and chaos for the horses, and thereby improve welfare. Driving aids were always used correctly. Most journeys started before noon, implying that the horses were often transported during the hottest moments of the day.

Still, on the one hand, interpretation of THI values is difficult because of a lack of reference framework for horses. On the other hand, not all parameters that affect thermal comfort are included in the THI. Therefore, it is not clear from what point on welfare is compromised. Stocking densities were not always according to relevant guidelines and significantly smaller horses or stallions were not always separated from the other horses. No influence of environmental parameters or transport characteristics on the degree of fighting behaviour could be demonstrated. On the contrary, the degree of aggression differed between the front and rear loading spaces of the same transport vehicle, suggesting that animal-specific factors, rather than environmental factors, determine the occurrence of aggressive behaviour. The willingness of all actors involved—slaughterhouses, transporters, loading crew, etc.—to conduct this study and to address shortcomings, underscores the growing awareness of animal welfare issues in Argentina and Uruguay.

## Figures and Tables

**Figure 1 animals-10-00602-f001:**
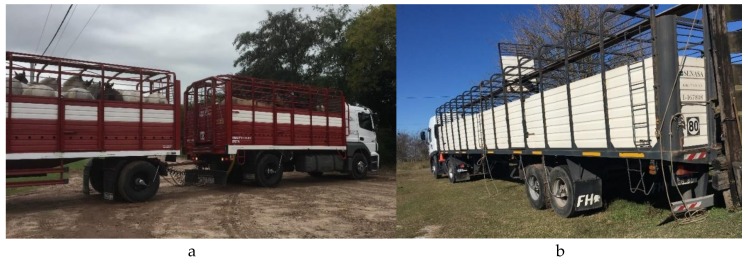
Example of different transport combinations. (**a**) A truck–trailer: a truck pulling a trailer and (**b**) A tractor–trailer: a pulling vehicle (without loading space) hauling a trailer through a fifth-wheel coupling.

**Figure 2 animals-10-00602-f002:**
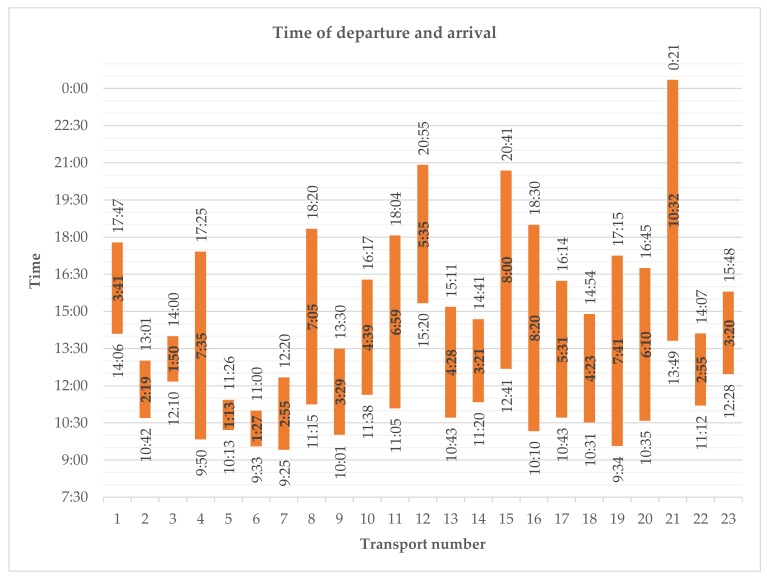
Arrival and departure times of the monitored transports. The duration of the transports is mentioned in the coloured bars. The transport numbers are the same as used in Table A3 and Table A4 in the Appendix A.

**Figure 3 animals-10-00602-f003:**
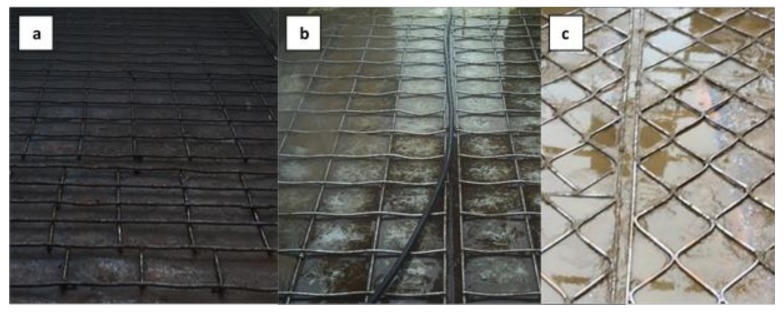
Different types of wire mesh that were used for the 23 observed slaughter horse transports. (**a**) Standard wire mesh—here addressed as “standard”; (**b**) standard wire mesh, which is curved between connection points—here called “curved”; (**c**) diamond shaped wire mesh, with connections in one plane—here called “diamond shaped”.

**Table 1 animals-10-00602-t001:** Dimensions and slope of the loading docks in the 18 collection points in degrees and in percentages.

Parameter	Average ± Standard Deviation	N	Minimum	Maximum	Median
Length (m)	4.01 ± 0.90	21	2.60	6.50	3.89
Height (m)	1.18 ± 0.17	21	0.93	1.74	1.17
Height side walls (m)	1.59 ± 0.22	19	1.16	1.94	1.60
Slope (°)	17.4 ± 3.6	21	9.3	23.6	17.1
Slope (%)	31.5 ± 7.0	21	16.4	43.6	30.7

**Table 2 animals-10-00602-t002:** Duration of loading and standstill after loading before departure.

Parameter	Average ± Standard Deviation	N	Minimum	Maximum	Median
Loading time (minutes)	12.2 ± 7.1	23	2.0	32.0	10.0
Loading time per horse (minutes/horse)	0.49 ± 0.27	23	0.19	1.23	0.42
Duration standstill before departure (minutes)	8.26 ± 4.28	23	2.00	19.00	8.00

**Table 3 animals-10-00602-t003:** The average, minimum and maximum temperature, RH and THI in the loading spaces. The average, minimum and maximum of each parameter was first calculated per transport and then per time period.

Parameter		Average ± Standard Deviation	N	Minimum	Maximum
Temperature		22.0 ± 5.0	38	6.0	35.5
	Spring	25.4 ± 3.7	11	15.0	35.5
	Summer	24.2 ± 2.2	11	17.5	32.5
	Autumn	15.3 ± 2.3	9	6.0	26.0
	Winter	21.7 ± 4.2	7	14.0	32.0
RH		57.4 ± 13.5	38	27.5	99.0
	Spring	52.6 ± 14.9	11	27.5	97.0
	Summer	59.9 ± 12.9	11	42.5	96.5
	Autumn	60.3 ± 6.2	9	43.5	80.0
	Winter	57.5 ± 16.4	7	31.5	99.0
THI		67.8 ± 6.5	37	45.5	83.0
	Spring	72.0 ± 4.6	11	59.0	83.0
	Summer	71.4 ± 2.7	11	62.6	81.5
	Autumn	59.1 ± 3.3	9	45.5	72.8
	Winter	66.5 ± 4.9	6	57.3	78.0

**Table 4 animals-10-00602-t004:** Correlations between temperature (T), relative humidity (RH) and temperature humidity (THI) in front and rear loading space. The *p*-value of “a” is the *p*-value of the correlation coefficient. T_AF_ = average temperature in the front loading space; T_AR_ = average temperature in the rear loading space; T_MinF_ = minimum temperature in the front loading space; T_MinR_ = minimum temperature rear loading space; T_MaxF_ = maximum temperature front loading space; T_MaxR_ = maximum temperature rear loading space. RH_AF_ = average RH front in front loading space; RH_AR_ = average RH in rear loading space; RH_MinF_ = minimum RH in front loading space; RH_MinR_ = minimum RH in rear loading space; RH_MaxF_ = maximum RH in front loading space; RH_MaxR_ = maximum RH in rear loading space. THI_AF_ = average THI in the front loading space; THI_AR_ = average THI in the front loading space; THI_MinF_ = minimum THI in the front loading space; THI_MinR_ = minimum THI in the rear loading space; THI_MaxF_ = maximum THI in the front loading space; THI_MaxR_ = maximum THI in the rear loading space.

Parameter	Equation (Y = aX + b)	N	r^2^	*p*-Value a	*p*-Value b
T_AF_ and T_AR_	T_AF_ = 1.01710 × T_AR_ − 0.26312	14	0.9847	<0.0001	0.7465
T_MinF_ and T_MinR_	T_MinF_ = 1.04965 × T_MinR_ − 0.77318	14	0.9840	<0.0001	0.3104
T_MaxF_ and T_MaxR_	T_MaxF_ = 0.85989 × T_MiaxR_ + 3.92441	14	0.8111	<0.0001	0.2244
RH_AF_ and RH_AR_	RH_AF_ = 1.00838 × RH_AR_ + 0.22294	14	0.9861	<0.0001	0.9020
RH_MinF_ and RH_MinR_	RH_MinF_ = 0.86383 × RH_MinR_ + 6.00346	14	0.9246	<0.0001	0.0668
RH_MaxF_ and RH_MaxR_	RH_MaxF_ = 0.99305 × RH_MiaxR_ + 0.78603	14	0.9676	<0.0001	0.8120
THI_AF_ and THI_AR_	THI_AF_ = 1.01061 × THI_AR_ − 0.64323	14	0.9878	<0.0001	0.7704
THI_MinF_ and THI_MinR_	THI_MinF_ = 1.04821 × THI_MinR_ − 3.03323	14	0.9831	<0.0001	0.2441
THI_MaxF_ and THI_MaxR_	THI_MaxF_ = 0.92326 × THI_MiaxR_ + 5.81791	14	0.8453	<0.0001	0.4859

**Table 5 animals-10-00602-t005:** Average available space per horse (m^2^/horse) and average density (kg/m^2^). T1 = front loading space; T2 = rear loading space.

Parameter		Average ± Standard Deviation	N	Minimum	Maximum	Median
Surface area of loading space (m^2^)		34.70 ± 9.50	23	13.50	41.91	39.45
Average density of full loading space (m^2^/horse)		1.40 ± 0.33	23	0.94	2.45	1.38
Density T1 (m^2^/horse)		1.40 ± 0.36	23	0.80	2.45	1.33
	Compartment 1	1.52 ± 0.53	8	1.03	2.68	1.46
	Compartment 2	1.38 ± 0.44	7	0.97	2.25	1.33
	Compartment 3	1.25 ± 0.35	2	1.00	1.50	1.25
Density T2 (m^2^/horse)		1.38 ± 0.22	15	0.99	1.88	1.38
	Compartment 1	1.36 ± 0.27	6	1.06	1.74	1.36
	Compartment 2	1.53 ± 0.49	6	1.15	2.48	1.37
	Compartment 3	1.86 ± 0.53	2	1.49	2.23	1.86
Average density (kg/m^2^)		308 ± 53	21	191	402	327
Density T1 (kg/m^2^)		312 ± 63	21	191	473	308
	Compartment 1	283 ± 65	6	175	352	304
	Compartment 2	322 ± 57	6	208	365	338
	Compartment 3	362	1	362	362	362
Density T2 (kg/m^2^)		309 ± 42	14	227	354	326
	Compartment 1	334 ± 74	5	253	446	344
	Compartment 2	291 ± 67	5	191	362	322
	Compartment 3	212	1	212	212	212

**Table 6 animals-10-00602-t006:** Average transport duration, distance and average number of stops and police checks per transport and per 100 km. Duration of loading, unloading and standstill after loading and before unloading.

Parameter	Average ± Standard Deviation	N	Minimum	Maximum	Median
Transport duration (minutes)	296 ± 150	23	73	632	268
Distance (km)	295 ± 177	23	37	700	250
Number of stops per transport	2.74 ± 1.33	23	0.0	5.0	3.0
Number of stops per 100 km	1.14 ± 0.90	23	0.00	4.35	1.04
Number of police checks per transport	0.87 ± 1.00	23	0.00	3.00	1.00
Number of police checks per 100 km	0.39 ± 0.58	23	0.00	1.97	0.25
Standstill between arrival and unloading (minutes)	14.8 ± 12.6	22	3.0	45.0	10.0
Duration of unloading (minutes)	12.5 ± 9.5	20	2.0	36.0	9.0
Duration of unloading per horse (minutes/horse)	0.473 ± 0.285	20	0.115	0.947	0.388

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
