# Peer review of "Horse Transport to Three South American Horse Slaughterhouses: A Descriptive Study"

_animals, 2020, doi:10.3390/ani10040602_

Round 1

Reviewer 1 Report

Congratulations of a very interesting project that includes an extensive array of results.  A lot of this information has not been previously presented, so thank you for that.  This reviewer has several suggestions to improve the readability of the manuscript (some English language style changes should be addressed by the editors, but as written the meanings could still be determined).

Lines 134 and 458 - slope should be described in degrees with the % slope put in parentheses as done in the discussion section 4.1.

Line 134 and Table 1 - the slope % is different.

Figure 3 is not necessary.

Line 184 - should be Table 3 (not Table 1).

Table 3 - This reviewer would prefer to see a table showing the minimum/maximum/average temperatures, humidity, and THI values, not just the equation results.  This would be helpful in understanding lines 199-209 also.

Line 212 - Should have the word "Appendix" as the first word before "Table A1."  (An additional comment about the Appendix tables follows that could change this suggestion.)

Tables A1 and A2 could be moved into the text (with appropriate renumbering). 

Lines 230-232 and Table 4 - On Lines 230 -232, the Title for Table 4 appears as part of the text.  It should be noted that on pages 8 and 9, Table 4 appears twice.  The first insertion is out of place.

Lines 274-276 - Where does this formula come from?  It unnecessary to the point being made - Table 5 says it all.

Line 324 - The authors make the statement that the horses hit their head because the trapdoors are too low.  Since the reader was not on scene, it is possible that the horses were rushed and because of that were carrying their heads higher than normal causing them to hit their heads.  This reviewer does not question that in either case the horses should not hit their heads, just that there could be more than one contributing factor.

Line 423 - This reviewer would suggest inserting a note to see Table 7 because the information is also there.

Because the authors are European, the EU requirements are discussed as a standard.  Since the article will be available worldwide, it would be beneficial to include standards from other countries, particularly Australia (also subject to hot temperatures).  

It is also interesting that the authors frequently refer to welfare, yet draw not conclusions or make recommendations about what their findings in that area are, other than drivers need to be better educated.  The temperature/humidity issues for the long hauls would be significant, but not so much if the trip was only a few hours.

Reviewer 2 Report

General comments: This is a very interesting study that covers a topic of great importance- the welfare of horses being shipped to slaughter. The research is descriptive instead of hypothesis driven, which seems to be the ideal situation for this type of research. Many measurements were determined and attention given to detail. There are some places throughout the manuscript where minor grammatical errors were made. I would suggest having the manuscript edited by an English language editor. My biggest comment is in regards to the discussion. It contains quite a bit of results, and re-stating results. I would suggest one of two things- either having a “Results and Discussion” section instead of just results and just discussion, or making sure that your discussion section does not re-hash the results. There are also two tables in the discussion and these need to be moved to the results and properly addressed.

Specific comments:

Line 92- “tractor-trailers and truck-trailers”

Line 165- should be “(± 3.86)%”

Line 174- define “trapdoor” in the methods

Line 184- Table 3 accidentally mislabeled as Table 1. Also this table needs more explaining. Most people just report the p-value for the correlation coefficient not the slope intercepts. Could you report the correlation coefficient significance?

Line 249-250- There is a grammar issue here. Typically, if you use the word “respectively” the things to which you are referencing are in the same sentence. For instance, “13, 15, and 11% of loading spaces of trucks, tractor-trailers, and truck tractors, had no fighting, respectively.” This statement is confusing without the context.

Line 340 “had previous negative experiences”

Line 345- defining drivers in the method would help with clarity.

Line 352-354- this section on environment is just kind of hanging here at the end of this paragraph. It doesn’t belong with the data discussed in that section.

Line 414- one or more stallions

Line 453- The conclusions are quite long. I would shorten this to one paragraph. This section includes results and discussion, which is not appropriate.

Reviewer 3 Report

This is an interesting work, with some good a novel information.

Introduction is long enough to contextualize the problem. Regarding Materials and Methods, it would be interesting to have some information about horses, like type (ponies, thoroughbred, working horses, etc) and an average weight. 

A lot of results are presented, but follow a clear order. Some minor details:

P4L133: after averages, I think the in is out of context.

P6L184: it should be Table 3, not 1.

P7L221 and 223: Figure 4 after Figure 4a and b must be erased.

P8: Table 4 appears two times, and some confusion in the text between both tables needs some rewriting.

P10 Unloading: The last sentence, fits better in the beginning of the paragraph.

Discussion:

P11L317: should be those of beef cattle

P11L331-334: do you have any information regarding training of drivers? some training in management, horse behavior or welfare? It is known that trining people help a lot with better management.

P11L347-348: also, meat quality could be compromised when electric prods are used.

P11L352-354: that paragraph should be in the environmental parameters section.

P12L402=402: Table 7...those are results, therefore must be in that section and not in the discussion part.

Are some local (Argentina and Uruguay) regulations regarding horse transport? Most of the discussion is based on European regulations, and nothing is mentioned considering that those were the countries were the study was performed. It would be interesting to include local regulations.

P14L450-452: maybe a little more discussion regarding unloading is needed, or maybe discus it with loading.

Conclusions: maybe shortened it a little with less results (there is no need to repeat some results). Also, maybe mention that training people involved in transportation of horses could increase or improve welfare?

P15L481-482: maximum permitted transport duration of 8 hours...is for both Argentina and Uruguay? or its based only in EU regulations? should be mentioned.

Round 2

Reviewer 2 Report

Thank you for making suggested revisions, the manuscript is clearer now.